# The structure of species discrimination signals across a primate radiation

Sandra Winters[1,2†]*, William L Allen[1,2,3], James P Higham[1,2]

[1]Department of Anthropology, New York University, New York, United States;
[2]New York Consortium in Evolutionary Primatology, New York, United States;
[3]Department of Biosciences, Swansea University, Wales, United Kingdom

**Abstract** Discriminating conspecifics from heterospecifics can help avoid costly interactions between closely related sympatric species. The guenons, a recent primate radiation, exhibit high degrees of sympatry and form multi-species groups. Guenons have species-specific colorful face patterns hypothesized to function in species discrimination. Here, we use a machine learning approach to identify face regions most essential for species classification across fifteen guenon species. We validate these computational results using experiments with live guenons, showing that facial traits critical for accurate classification influence selective attention toward con- and heterospecific faces. Our results suggest variability among guenon species in reliance on single-trait-based versus holistic facial characteristics for species discrimination, with behavioral responses and computational results indicating variation from single-trait to whole-face patterns. Our study supports a role for guenon face patterns in species discrimination, and shows how complex signals can be informative about differences between species across a speciose and highly sympatric radiation.

***For correspondence:**
sandra.winters@bristol.ac.uk

**Present address:** [†]School of Biological Sciences, University of Bristol, Bristol, United Kingdom

**Competing interests:** The authors declare that no competing interests exist.

## Introduction

Closely related species living in sympatry face a potential challenge in discriminating between conspecifics and heterospecifics. Such decision-making has important selective outcomes, particularly in behaviors such as mate choice, with individuals choosing heterospecific mates often incurring substantial fitness costs (*Coyne and Orr, 2004*; *Gröning and Hochkirch, 2008*). One mechanism for avoiding the costs of interacting with heterospecifics is the use of species-specific signals that structure behavioral interactions between species. For instance, mating signals and associated mating preferences that differ between sympatric heterospecifics can function to maintain reproductive isolation across species boundaries (*Andersson, 1994*). Such signals are predicted to be salient and distinctive (*Dawkins and Guilford, 1997*), with sympatric species under selective pressure to diversify. A pattern in which signal distinctiveness increases with degree of sympatry, known as character displacement (*Brown and Wilson, 1956*; *Pfennig and Pfennig, 2009*), has been observed in a wide variety of animal groups (insects: *Higgie et al., 2000*; *Lukhtanov et al., 2005*; *Stanger-Hall and Lloyd, 2015*; birds: *McNaught and Owens, 2002*; gastropods: *Kameda et al., 2009*; mammals: *Allen et al., 2014*; anurans: *Pfennig and Rice, 2014*; *Gordon et al., 2017*). Importantly, signals that function to promote behavioral interactions with conspecifics should elicit increased interest from conspecifics compared to heterospecifics (*Mendelson and Shaw, 2012*).

Species in evolutionarily young animal radiations may be at particular risk of hybridization and other costly interactions with heterospecifics due to behavioral similarities and a lack of post-mating barriers to reproduction (*Mallet, 2005*). One such radiation is the guenons (tribe Cercopithecini), a group of African primates consisting of 25–38 recognized species (*Grubb et al., 2003*; *Groves, 2005*; *Lo Bianco et al., 2017*) that diverged from papionin primates around 11.5 million years ago (*Tosi et al., 2005*; *Guschanski et al., 2013*). Guenons exhibit high degrees of sympatry

and often form polyspecific groups in which multiple species travel and forage together (*Jaffe and Isbell, 2011*). Many guenons therefore interact with heterospecifics that share general patterns of morphology (e.g. overall body size/shape) and behavior (e.g. activity patterns). In such circumstances, discriminating between con- and heterospecifics may be particularly important, especially in a mating context. Hybridization between sympatric guenon species has been observed in captive and wild settings but is rare in most natural circumstances (*Detwiler et al., 2005*), suggesting the existence of barriers to heterospecific mating within mixed-species groups.

Guenons are among the most colorful and visually patterned groups of primates, with many species exhibiting extraordinary and unique face markings (*Kingdon, 1980*; *Kingdon, 1988*; *Kingdon, 1997*; *Bradley and Mundy, 2008*; *Allen et al., 2014*) that cannot be reliably computationally classified by sex across all guenon species (*Kingdon, 1997*; *Allen and Higham, 2015*). Kingdon hypothesized that guenons use their divergent facial appearances to distinguish between species and select appropriate mates (*Kingdon, 1980*; *Kingdon, 1988*; *Kingdon, 1997*). This young and impressively diverse primate radiation represents a fascinating test case of how visual signals are involved in species radiations and mixed-species interactions (*Pfennig and Pfennig, 2009*; *Pfennig and Pfennig, 2010*; *Pfennig, 2016*; *Stuart et al., 2017*). Recent empirical work has begun to generate evidence for a key role for face patterns in guenon phenotypic and species diversification. Images of guenon faces can be reliably classified by species using computer algorithms (*Allen et al., 2014*; *Allen and Higham, 2015*), demonstrating that guenon faces exhibit species-specific patterns that are potentially informative and biologically relevant to receivers. Guenon face patterns also exhibit character displacement, with facial distinctiveness between species increasing with degree of geographic range overlap across the group (*Allen et al., 2014*). Moreover, facial components common across species (nose spots and eyebrow patches) alone can be used to computationally classify species even when segmented from the rest of the face (*Allen and Higham, 2015*). This suggests that guenon faces may be somewhat modular, with particular face regions being potentially informative regarding species identity. Which face regions are most important, and the extent to which such regions vary across species, remains an open question that is of key significance to understanding how complex signals involved in species discrimination evolve. Critically, it is unknown whether variation across guenon species in purported species discrimination signals is perceived and acted on by con- and heterospecific receivers.

Here, we use a machine learning approach to identify guenon face regions that are most important for correct species classification by a computer. These results objectively identify the signal components most likely to be useful to guenon receivers. We use them to determine which signal properties to systematically investigate in behavioral experiments with guenon observers. The machine-learning stage is critical, as many experiments that investigate behavioral responses to complex signals select manipulations based on the perceptions of investigators, which introduces anthropocentric bias (*Patricelli and Hebets, 2016*). Using the guenon face image database produced by *Allen et al. (2014)*, we couple eigenface decomposition of guenon faces (*Turk and Pentland, 1991*) with a novel occlude-reclassify scheme in which we systematically block each part of the face and reclassify the image. This allows us to document the spatial distribution of species-typical information across guenon faces by identifying which face regions, when obscured, cause the break-down of correct species classification. Eigenface decomposition was originally developed for individual face discrimination in humans (*Turk and Pentland, 1991*); feature detection based on eigenfaces is also applicable to other types of discrimination tasks involving complex animal signals (*Weeks et al., 1999*; *Shiau et al., 2012*; *Perera and Collins, 2015*) and has been used previously to quantify guenon facial variation (*Allen et al., 2014*). The type of perceptual face space generated by eigenface decomposition has correlates in mammalian visual processing systems (*Leopold et al., 2001*; *Cowen et al., 2014*; *Chang and Tsao, 2017*), lending biological credibility to this approach.

We ran occlude-reclassify analyses using both the entire set of fifteen guenon species as well as two subsets, which included species with geographic ranges that overlap those of two guenon species. The full analysis including all species allows us to identify facial traits that differentiate each species from others in the tribe, and these results are therefore broadly applicable across species. Additional subset analyses represent a more biologically realistic discrimination task for the two species that were the focus of experiments; here, the key results are for the two species that form the basis for these subsets.

After identifying the face regions that cause a break-down in classification, and thus those that should be important for correct species identification, we then presented captive putty nosed monkeys (*Cercopithecus nictitans*) and mona monkeys (*C. mona*) with images of con- and heterospecific faces in a behavioral discrimination task measuring the looking time toward each stimulus. Ours is the first direct measure of guenon responses to con- and heterospecific faces, which is crucial for clarifying the biological relevance of guenon face patterns and for validating previous correlational results. Looking time experiments are commonly used in studies of human infants and non-human animals to gauge visual attention and interest, and in so doing to infer underlying perceptual and cognitive abilities and biases (reviewed in *Winters et al., 2015*). Our looking time experiments are not designed to assess the neurological mechanisms underlying object detection or classification, but rather to assess the level of interest elicited by different classes of stimuli and to identify the facial features that contribute to this interest. Differences in looking time between classes of stimuli can be difficult to interpret due to various and often unpredictable novelty and familiarity effects (*Winters et al., 2015*), however primates reliably exhibit a visual bias (i.e. greater looking time) toward images of conspecifics compared to those of heterospecifics (*Fujita, 1987*; *Demaria and Thierry, 1988*; *Dufour et al., 2006*; *Méary et al., 2014*; *Rakotonirina et al., 2018*). Our experimental trials involve the simultaneous presentation of paired con- and heterospecific faces, with trial stimuli varying in facial pattern components across experimental conditions. In this approach, a statistical difference in looking time between classes of stimuli is a behavioral indicator of an ability to discriminate between these classes as well as an indicator of relative interest in the two classes. Experimental stimuli are designed to vary in the presence or absence of nose spots for putty nosed monkeys and eyebrow patches for mona monkeys, on the basis that each of these features is within the region of the face identified by our machine learning approach as being critical for that species. In our first experimental condition, subjects are presented with the face of a conspecific and that of a heterospecific which does not display the relevant focal face trait. This condition represents a baseline in which there is no conflation between species and species-typical face traits, and confirms that the general pattern of visual biases for conspecific faces holds for the individuals in the two guenon species tested here. It also represents a biologically realistic comparison, as guenons are typically surrounded by heterospecifics with faces dissimilar to their own (due to character displacement, *Allen et al., 2014*). A conspecific bias in this condition would suggest that subjects find this class of stimuli more interesting, and may translate into behavioral differences in interactions with conspecifics relative to heterospecifics. In our subsequent experimental conditions, we present stimuli in which species and focal face traits are conflated: in condition two, we present a conspecific face paired with a heterospecific face that shares the facial trait; in condition three, we present a conspecific face modified to remove the facial trait paired with a heterospecific face that shares the facial trait (for example stimuli, see *Figure 1*). These latter trials therefore ask the question: how is a conspecific bias affected by manipulations of which facial pattern components are available? The stimuli in these trials do not mimic natural scenarios, but including artificial stimuli such as modified conspecific faces allows us to evaluate the role of these face traits in directing guenon visual attention. We present results analyzing visual biases in the first condition, as well as a unified analysis of visual biases across all experimental conditions to identify the overall drivers of looking time across conditions. This approach allows us to assess generalized species biases in degree of interest as well as the extent to which particular face regions influence these biases.

We predicted variability across species in the face regions identified by our occlude-reclassify procedure, but made no predictions regarding which regions in particular would be essential for each species. In looking time experiments, we predicted that putty nosed and mona monkeys would exhibit visual biases toward face images of conspecifics, and that these biases would be influenced by species-typical facial characteristics identified as important for correct species classification. Such a pattern of results is consistent with a role for species discrimination signals in facilitating interspecific interactions, such as maintaining reproductive isolation via mate choice, and in generating and maintaining phenotypic variation in one of the most speciose and diverse primate radiations. Species discrimination requires differentiating between conspecifics and all heterospecifics at the very least, however it may be possible that animals are also able to discriminate between different heterospecific species. Ultimately, by examining how aspects of highly complex signals encode species identity and influence receiver biases, this research increases our understanding of how selection for species identity signaling generates phenotypic diversity.

**Figure 1.** Example experimental stimulus pairs. Subjects were shown a pair of stimulus images consisting of a conspecific and a heterospecific. Facial traits (nose spots for putty nosed monkeys and eyebrow patches for mona monkeys) were varied across trials, with conspecifics paired with a heterospecific species that shares the facial trait (row 1) and one that does not (rows 2 and 3). Conspecifics were displayed either naturally (rows 1 and 2) or with the facial trait removed (row 3). All subjects participated in all three trial types. Trial order and stimulus image side were counterbalanced across subjects.

The online version of this article includes the following figure supplement(s) for figure 1:

**Figure supplement 1.** The experimental apparatus.

## Results

### Occlude-reclassify machine classification

We began by confirming that guenons could be reliably classified by species based on eigenface decomposition (*Allen et al., 2014*). In the full dataset, average subject images were correctly classified by species 99.31% of the time, and distinct images were correctly classified 93.03% of the time. All correctly classified images (n = 654) were used to identify face regions of critical importance to correct species classification by the computer algorithm, using our occlude-reclassify scheme. We identified essential face regions in all guenon species that, when occluded, led to incorrect species classification (*Figure 2*; for full resolution images see *Supplementary file 1*). Species differed in the importance of different face regions as well as the extent to which important regions were concentrated in specific facial features or were more widely distributed across larger face areas (*Figure 3*). For example, the nose spot of the putty nosed monkey was the most critical facial feature identified across all species. The putty nosed monkey had the highest mean error rate for misclassified face

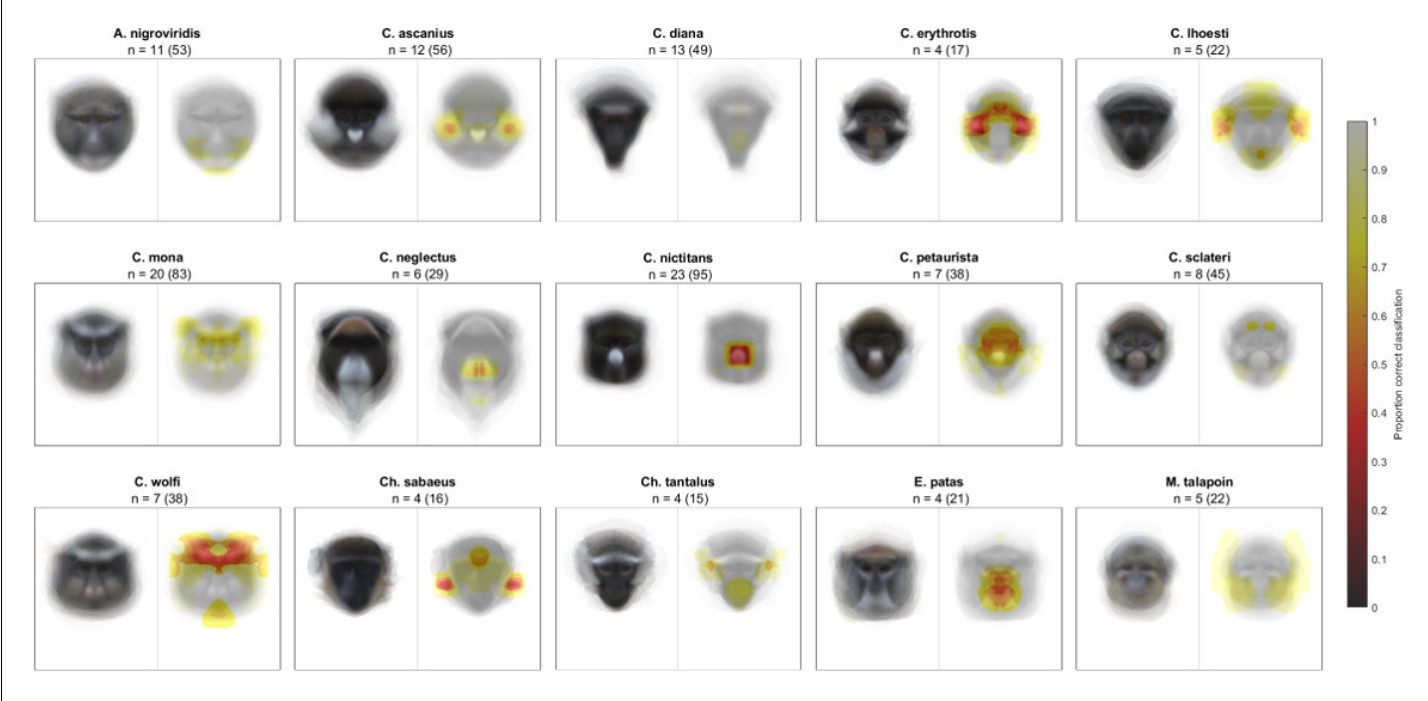

**Figure 2.** Likelihood of correct classification based on occlusion of different face regions. Species average faces are displayed on the left and heatmaps identifying critical face regions on the right. Sample size is reported as n = number of individuals (number of total images).
The online version of this article includes the following figure supplement(s) for figure 2:

**Figure supplement 1.** Likelihood of correct classification based on occlusion of different face regions with the occluder set to the species-specific mean face color and run on the left hemi-face (see Appendix), using a subset of species that overlap in range with putty nosed monkeys.

**Figure supplement 2.** Likelihood of correct classification based on occlusion of different face regions with the occluder set to the species-specific mean face color and run on the right hemi-face (see Appendix), using a subset of species that overlap in range with putty nosed monkeys.

**Figure supplement 3.** Likelihood of correct classification based on occlusion of different face regions using an average gray occluder on left hemi-face (see Appendix), using a subset of species that overlap in range with putty nosed monkeys.

**Figure supplement 4.** Likelihood of correct classification based on occlusion of different face regions using an average gray occluder on right hemi-face (see Appendix), using a subset of species that overlap in range with putty nosed monkeys.

**Figure supplement 5.** Likelihood of correct classification based on occlusion of different face regions with the occluder set to the species-specific mean face color and run on the left hemi-face (see Appendix), using a subset of species that overlap in range with mona monkeys.

**Figure supplement 6.** Likelihood of correct classification based on occlusion of different face regions with the occluder set to the species-specific mean face color and run on the right hemi-face (see Appendix), using a subset of species that overlap in range with mona monkeys.

**Figure supplement 7.** Likelihood of correct classification based on occlusion of different face regions using an average gray occluder on left hemi-face (see Appendix), using a subset of species that overlap in range with mona monkeys.

**Figure supplement 8.** Likelihood of correct classification based on occlusion of different face regions using an average gray occluder on right hemi-face (see Appendix), using a subset of species that overlap in range with mona monkeys.

**Figure supplement 9.** Likelihood of correct classification based on occlusion of different face regions using an average gray occluder on left hemi-faces (see Appendix).

**Figure supplement 10.** Likelihood of correct classification based on occlusion of different face regions using an average gray occluder on right hemi-faces (see Appendix).

**Figure supplement 11.** Likelihood of correct classification based on occlusion of different face regions with the occluder set to the species-specific mean face color and run on the right hemi-face (see Appendix).

regions – indicating that the face regions identified had the highest likelihood of causing misclassification when occluded – with the essential regions centered exclusively on the nose. Thus, in the putty nosed monkey the nose is the only essential face feature; when the nose is occluded species classification breaks down, whereas occluding any other face region has no effect. In contrast, in other species our classifier relied on broader regions of the face, with larger face regions identified as important for correct classification and the classifier relying less exclusively on a single feature. The mona monkey is a good example of this, with disparate face regions including the cheeks,

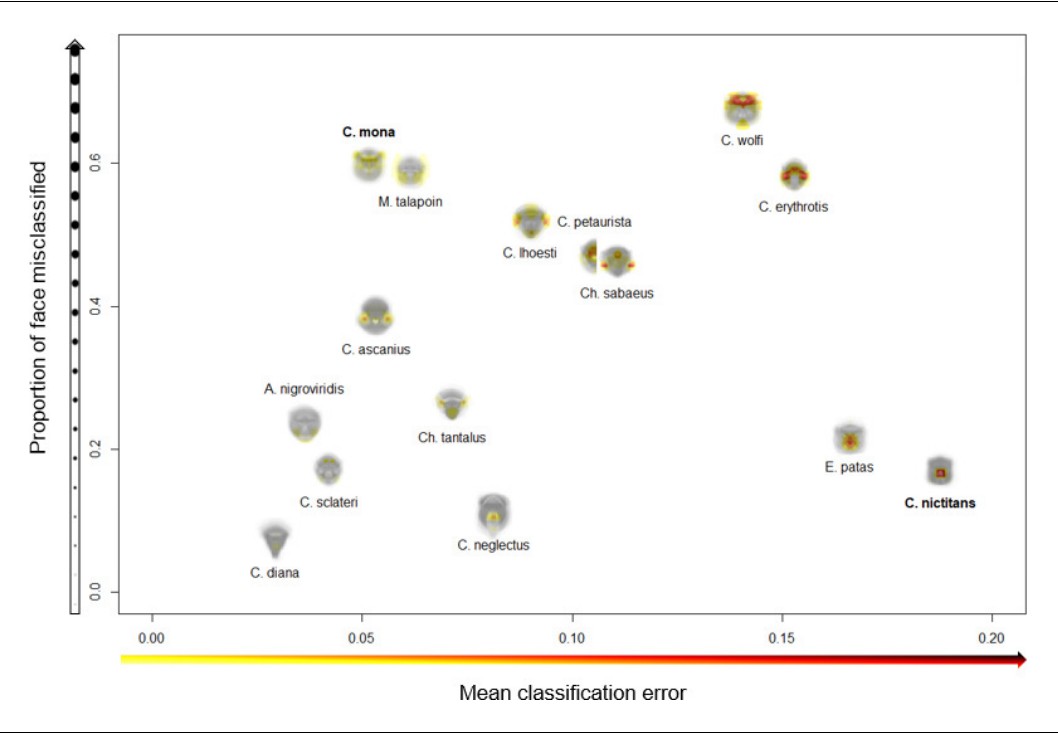

**Figure 3.** Variation across species in face regions identified as essential for correct species classification. The proportion of the face misclassified (y-axis) indicates the spread of essential regions across the face; higher values signify broader spread and lower values more concentrated regions. The mean classification error (x-axis) measures the relative importance of identified features; higher values indicate higher rates of misclassification, suggesting identified regions are particularly essential for correct species classification. Experimental results are presented for *C. mona* and *C. nictitans* (*Figure 4*).

eyebrows, and ear tufts all influencing correct classification of this species. In some species negative space is important, suggesting that what makes the faces of some species distinctive may be the absence of certain facial traits. For instance, in *M. talapoin* the absence of distinctive traits along the sides of the face – such as cheek and/or ear tufts observed in other species – appears to be important.

Results generated using subsets of species that have range overlap with putty nosed monkeys and mona monkeys were similar to those generated using the entire image set. In particular, the critical facial regions identified when comparing the putty nosed monkey with sympatric species also included the nose spot as the only essential feature (*Figure 2—figure supplements 4–7*). Analyses run on left and right hemi-faces differed somewhat in how important different face regions were for correct classification, but taken as a whole still show that a broader face region including eyebrow patches is important for correct classification when classifying mona monkeys with respect to sympatric species (*Figure 2—figure supplements 8–11*).

## Looking time experiments

Our experiments presenting subjects with pairs of con- and heterospecific faces revealed visual biases in resulting eye gaze in both putty nosed and mona monkeys. Putty nosed and mona monkeys viewed images for a mean (± standard deviation) of 3.89 ± 1.98 seconds (s) and 4.58 ± 2.52 s across all trials, respectively. In the subset of trials that included a natural conspecific and a heterospecific without the relevant face trait (i.e. those where the relevant facial traits are not spread across both con- and heterospecific faces), species (and therefore also facial trait) was a significant predictor of looking behavior (putty nosed monkeys: mean looking time at conspecifics = 2.08 ± 1.40 s, at heterospecifics = 1.64 ± 1.15 s; Chisq = 63.312, p<0.001; mona monkeys: mean looking time at conspecifics = 2.41 ± 1.48 s, at heterospecifics = 2.03 ± 1.06 s; Chisq = 30.755, p<0.001), with both

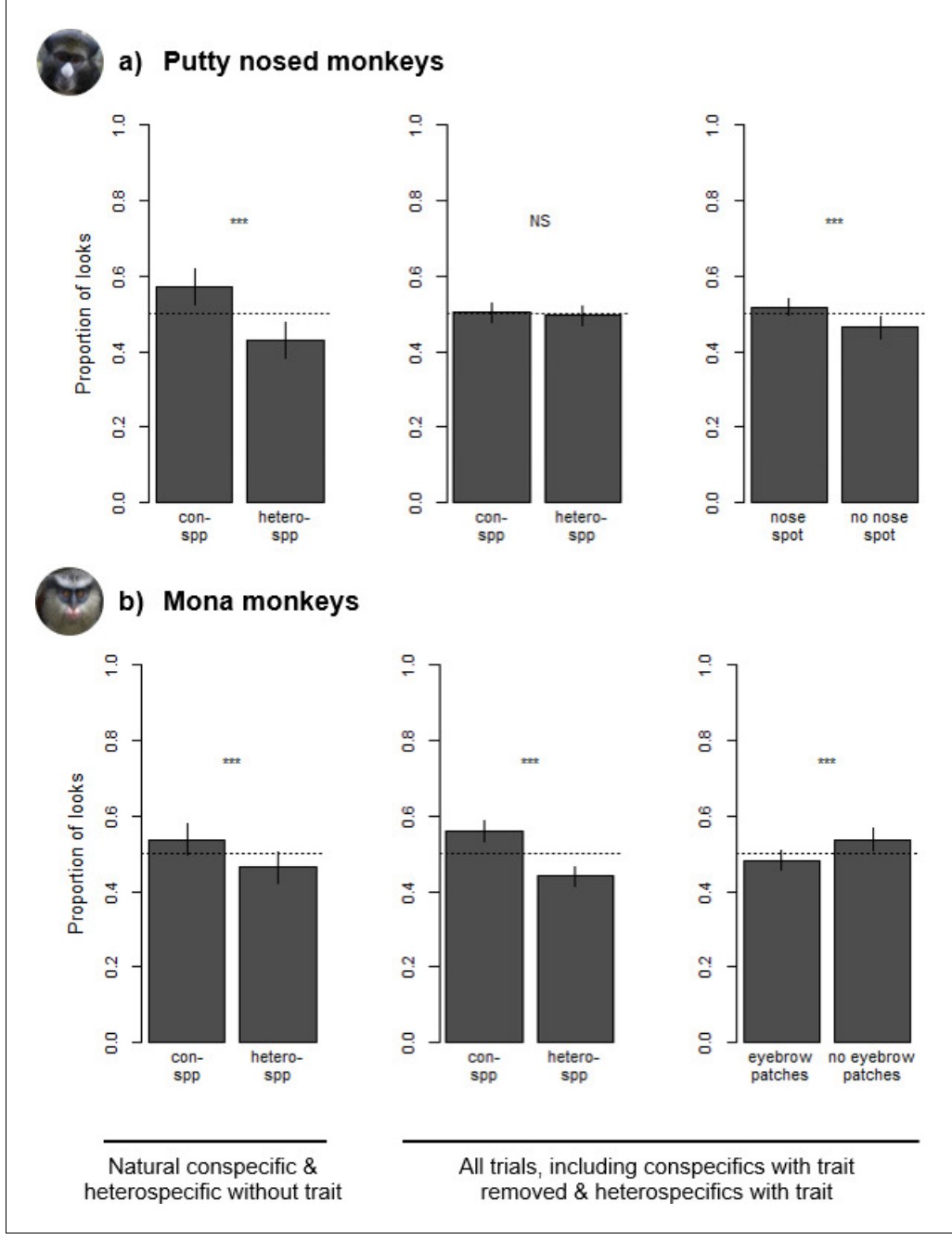

**Figure 4.** Species and trait biases observed during looking time tasks with (a) putty nosed monkeys and (b) mona monkeys. Each trial involved the simultaneous presentation of two images, with proportion of looks calculated as the relative duration of eye gaze at each image. Leftmost plots depict differences in looking time in trials consisting of conspecifics and heterospecifics without the relevant facial trait. Center and right plots depict looking time differences across all trials – which also include heterospecifics with the relevant facial trait and conspecifics without it – with species biases depicted in the center and trait biases on the right. Results are based on 18 putty nosed monkeys and 16 mona monkeys. Each subject participated in three trials (see *Figure 1* for example stimuli for each trial type). Error bars indicate the standard error of the mean.

putty nosed and mona monkeys exhibiting a conspecific bias (respectively: z = 7.920, p<0.001, conditional odds ratio = 1.66; z = 5.536, p<0.001, conditional odds ratio = 1.41; *Figure 4*). Odds ratios compare the likelihood of events, which can be informative for evaluating effect size; here, for every second spent looking at heterospecific faces putty nosed monkeys spent 1.66 s and mona monkeys 1.41 s looking at conspecific faces when all other variables are held constant.

Across all trials, in putty nosed monkeys model comparisons revealed that looking behavior was significantly influenced by facial trait (nose spot v. no nose spot; mean looking time at faces with nose spots = 1.99 ± 1.19 s, at faces without nose spots = 1.86 ± 1.34 s; Chisq = 11.511, p<0.001) and image location (mean looking time at left image = 1.87 ± 1.21 s, at right image = 2.02 ± 1.27 s; Chisq = 18.065, p<0.001), but not by species (mean looking time at conspecifics = 1.97 ± 1.33 s, at heterospecifics = 1.91 ± 1.15 s; Chisq = 3.055, p=0.081). Overall, putty nosed monkeys looked longer at stimulus faces that displayed a white nose patch (z = 3.343, p<0.001, conditional odds ratio = 1.14; *Figure 4*), their diagnostic species trait, regardless of species identity. Putty nosed monkeys also exhibited a significant right gaze bias (z = 4.289, conditional odds ratio = 1.17, p<0.001). None of the other variables relating to subject, stimulus, or trial characteristics were statistically significant (all p>0.1; *Supplementary file 2*).

In mona monkeys, model comparisons revealed that looking behavior was significantly influenced by species (mean looking time at conspecifics = 2.56 ± 1.83 s, at heterospecifics = 2.02 ± 1.24 s; Chisq = 177.480, p<0.001), facial trait (mean looking time at faces with eyebrow patches = 2.22 ± 1.67 s, at faces without eyebrow patches = 2.44 ± 1.39 s; Chisq = 29.462, p<0.001) and a species*trait interaction (Chisq = 8.242, p=0.004). Across all trials, mona monkeys looked longer at conspecifics (z = 9.945, conditional odds ratio = 1.84, p<0.001; *Figure 4*) and at faces without white eyebrow patches, which are one component of their broader diagnostic species trait (z = 5.851, conditional odds ratio = 1.36, p<0.001). There was also an interaction between these two variables, with mona monkeys looking longer at heterospecific faces with white eyebrow patches (z = 2.868, conditional odds ratio = 1.24, p=0.004). None of the other variables relating to subject, stimulus, or trial characteristics played a significant role in mona monkey visual biases (all p>0.1; *Supplementary file 2*).

## Discussion

Our machine classifier results identified certain face regions as being critical to correct species classification across the guenons, and experiments undertaken for two target species provided results that were consistent with the results of the classifier. This convergence of results using disparate methods reinforces the validity of both, and ties computationally derived results directly to guenon perception, demonstrating the utility of machine learning for identifying biologically relevant signal components. To our knowledge, ours is the first analysis to use machine classification combined with the systematic occlusion of image regions to characterize the relevant signaling information encoded in an animal's appearance. This approach, based on research in the field of computer vision designed to assess the contribution of image contents to object classification (*Zeiler and Fergus, 2014*), is useful for objectively quantifying the relative roles of different signal components with respect to overall signal function. In closely-related sympatric species, selection against mating or other behavioral interactions with heterospecifics is often associated with the evolution of species-typical traits used to maintain reproductive and behavioral isolation. The guenons, a recent and diverse radiation that exhibit mixed species groups in which hybridization is rarely observed, exemplify this phenomenon. By showing how species classification is dependent on different aspects of face patterning and that this links with looking time toward con- and heterospecifics, our analyses support a role for guenon face patterns in species discrimination, and identify specific face regions potentially critical for this function. This parsing of critical signal components is essential for understanding the phenotypic evolution of complex signals and identifying relevant axes of signal variation for additional analyses.

Our occlude-reclassify analysis identified face regions critical to correct species classification by a machine classifier in all guenon species included in our study. Critical regions differed in both location and spread across the face, suggesting variation in potential use across species. For some guenons, reliance on a single facial characteristic may be sufficient for species discrimination. The best example of this in our data set is the putty nosed monkey, where our machine classifier relied

exclusively on the white nose spot to classify this species. That is, occlusion of any other region resulted in correct classification, but when the nose spot was occluded classification failed. This result is reinforced by our experiments, in which putty nosed monkey visual attention was driven wholly by the presence of nose spots. Putty nosed monkeys exhibited a conspecific bias when presented with natural con- and heterospecific faces, as is typical in primates, however including stimuli depicting heterospecifics with nose spots and conspecifics without nose spots completely obscured this conspecific bias. This combination of results illustrates the importance of nose spots in this species. It is worth noting that putty nosed monkey nose spots are the most straightforward facial trait documented in our analysis (i.e. putty nosed monkeys were only misclassified when the nose spot was occluded and occluding the nose spot led to a high rate of misclassification) and that putty nosed monkeys are the only species in our analyses for which the classifier identified only one critical face region; the relative simplicity of the face and related visual biases in this species is likely exceptional. On the whole, species discrimination signals in a large radiation with varying patterns of sympatry are expected to be complex and multidimensional, and it is likely that only some species can exhibit single-trait-based signals and visual biases without the system breaking down. This is supported by our results showing that for all other guenon species our classifier identified multiple face regions that were critical for species discrimination.

Not all guenons exhibited critical face regions restricted to isolated facial traits, and our machine classifier sometimes relied on disparate face regions. In our data set, the mona monkey is a good example of such a species. As in putty nosed monkeys, our experiments with mona monkeys supported these computational results. Mona monkeys exhibited a conspecific bias across all trials, regardless of single trait manipulations, as well as an additional bias based on the presence of eyebrow patches. Thus, eyebrow patches alone do not appear to be the sole driver of visual attention in mona monkeys. We predict that additional manipulation of other face regions would be necessary to redirect their visual attention. Nonetheless, that mona monkey attention is still influenced by this species-typical trait shows that it is important but not essential, a result predicted by our computational analyses. It is unclear why mona monkeys would look longer at stimuli without eyebrow patches; however, it is possible that utilization of the whole face causes increased attention to incongruency (e.g. conspecifics without eyebrow patches or heterospecifics with them). Another possibility is that this manipulation resulted in odd-looking faces that drew subjects' attention, although the absence of eyebrow patches in most guenon species would suggest that these features are not necessary for general face processing. Our results suggest that in mona monkeys, species discrimination may be based on broader face information, and the perceptual processes involved in assessing potential mates could be similar to generalized holistic face processing mechanisms observed in other primates (*Dahl et al., 2009*). Complex multicomponent signals also have a greater potential for signal redundancy (*Partan and Marler, 1999*; *Hebets and Papaj, 2005*; *Higham and Hebets, 2013*), although our results from mona monkeys showing that a single component of a complex signal does not drive attention on its own suggest that redundant face regions are more likely to represent enhancement (i.e. an increased response to the combined signal compared to each component) than equivalence (i.e. the same response to each component or the combined signal) of alternative regions (*Partan and Marler, 1999*).

Our results suggest that guenons, while united by a general pattern of facial diversification and the probable use of faces in mate choice, may vary across species in the specific traits and processes that are involved in discriminating between conspecifics and heterospecifics. Our pattern of experimental results for putty nosed monkey nose spots and mona monkey eyebrow patches is interesting because we know that both traits are sufficiently informative to allow discrimination between species that share these features (*Allen and Higham, 2015*), yet they influence attention differently in the two species. This disparity highlights the importance of testing receiver perception directly, and also indicates the importance of future experimental work on live guenons in additional species. The fact that our experimental results with the two species of guenons tested in the present study line up with predictions generated by our occlude-reclassify analysis nonetheless implies that these computationally derived results may be biologically valid across the radiation. While the observed differences in looking time were small, the magnitude of these differences is similar to those reported in looking time studies of humans (*Shimojo et al., 2003*) and other non-human primates (*Dubuc et al., 2014*; *Dubuc et al., 2016*), in which phenotypes that elicit increased visual gaze have also been associated with reported judgements of attractiveness and behavioral indicators of mating interest,

respectively. This suggests that these differences are biologically meaningful and may influence guenon decision making. Here, our results strictly show that visual attention in guenons is influenced by variation in face pattern components; the extent to which these attentional biases translate to other behavioral contexts awaits additional investigation.

Interestingly, we found no sex differences in visual biases for either species, suggesting that selective pressures on species discrimination signaling and visual preference traits are similar between sexes. Most guenons live in polygynous mating systems (one male, multiple females) and males exhibit minimal paternal care. This could theoretically lead to the evolution of increased choosiness and discriminative capabilities in females, if interspecies interactions such as hybridization are more costly to them. However, many guenon groups experience seasonal influxes of males during the mating season; these extra-group males tend to be attractive to females, and the group-male is often unable to competitively exclude them (*Cords, 2012*; *Cords, 2004*). In such circumstances, group-males that opt to protect reproductive access to heterospecifics would incur substantial fitness costs, and males in mixed-species groups (as are typical for most guenons: *Jaffe and Isbell, 2011*) could gain fitness advantages by conceding reproductive access to heterospecific females while preferentially mate guarding conspecifics. As such, we predict that engagement with heterospecific partners is costly in both sexes. Field observations documenting the roles of males and females in initiating and preventing interactions between species would help to evaluate this prediction and contextualize our results.

In guenons, an observed lack of hybrids in most polyspecific groups (*Detwiler et al., 2005*) is notable given that hybridization is known to be possible between many guenon species (*Kingdon, 1980*; *Kingdon, 1997*; *Detwiler et al., 2005*) and indicates the existence of pre-mating barriers to reproduction. Increased eye gaze is associated with increased mating interest in humans (*Shimojo et al., 2003*) and non-human primates (*Dubuc et al., 2014*; *Dubuc et al., 2016*), suggesting that our experimental results could be indicative of an increased interest in mating with conspecifics based on their face patterns in our putty nosed and mona monkey subjects. Combined with previous work (*Kingdon, 1980*; *Kingdon, 1988*; *Kingdon, 1997*; *Allen et al., 2014*; *Allen and Higham, 2015*), our results support the hypothesis that guenon face patterns play a role in mate choice and reproductive isolation in this group. However, it remains possible that the selection pressure for species discrimination traits in guenons arises partially or entirely from other functions where behavioral coordination or avoidance between species is advantageous, such as in foraging decisions (*Kingdon, 1997*; *Jaffe and Isbell, 2011*). Careful field observations would be needed to distinguish between such possibilities.

Our occlude-reclassify approach is a novel method for identifying the distribution of information in complex signals and can be used for any question that can be conceptualized as a discrimination problem and analyzed using machine classification. This method therefore has broad utility within sensory ecology and could help to better understand the link between form and function in the complex signals that are common in many animal groups. The objectivity of the approach is important, as it allows researchers to intelligently targetspecific signal components for further analysis without reference to their own perceptions of their salience. This is particularly important when studying species with sensory and perceptual systems very different from our own (*Endler, 1990*; *Bennett et al., 1994*). Where possible, combining this approach with a biologically realistic encoding scheme, such as classification within a perceptual face space based on eigenface scores (*Leopold et al., 2001*; *Cowen et al., 2014*; *Chang and Tsao, 2017*) as used here, increases the biological validity of results.

Our research broadens our understanding of how morphology and social decision-making can interact to structure interactions between species living in sympatry. In guenons, facial features like white nose spots are highly salient, attention-grabbing, and distinctive, and our combined results suggest that these may be important for species discrimination. Guenon behavioral repertoires, such as nose-to-nose touching observed in wild putty nosed monkeys (SW, personal observation) and red-tailed monkeys (*Estes, 1991*), further reflect the importance and biological relevance of these traits. Primates preferentially attend to facial information (*Tsao et al., 2008*; *Kano and Tomonaga, 2009*), making face patterns particularly suited to influencing behavior and decision-making in con- and heterospecifics. Interestingly, multiple other primate groups are also characterized by rapid speciation, high degrees of sympatry, and diverse facial appearances (e.g. callitrichids, *Opazo et al., 2006*; gibbons, *Carbone et al., 2014*), and comparative analyses have linked face pattern

complexity to number of sympatric congenerics in primates (*Santana et al., 2012*; *Santana et al., 2013*), suggesting that selection for species discrimination may be a key driver of diversity in this group. Our work linking signal appearance to visual attention in a recent and diverse primate radiation highlights how such evolutionary processes can be important in generating animal phenotypes.

## Materials and methods

### Identifying critical face regions

#### Image collection and processing

Guenon face pattern analyses are based on an existing database of guenon face images from 22 guenon species (*Allen et al., 2014*). Detailed methods of image collection and processing have been published previously (*Allen et al., 2014*). Briefly, we used digital images of captive guenons collected using a color-calibrated camera. Multiple images were taken of each subject while in a front-facing position under indirect light. Images were transformed from camera RGB color space to guenon LMS color space, defined by the peak spectral sensitivities of guenon long, medium, and short wavelength photoreceptors (*Bowmaker et al., 1991*; *Westland and Ripamonti, 2004*; *Stevens et al., 2007*). All images were then standardized with respect to illumination, size, blur, and background. Each image was resized to be 392 by 297 pixels. All pixel values were represented using double-level precision.

To avoid classifying species based on a very small number of exemplars, we restricted our analyses to species represented by at least four individuals (range: 4–23) in our image database (i.e. all classifications in a leave-one-out procedure are made based on at least three exemplars; see below). Our analysis is therefore based on 599 total images of 133 individuals, collectively representing 15 guenon species (for species-specific sample sizes, see *Figure 2*).

#### Occlude-reclassify procedure

Guenon face images can be reliably classified by species based on eigenface features (*Allen et al., 2014*; *Allen and Higham, 2015*). This approach relies on dimensionality reduction via principal component analysis (PCA) to extract relevant features from face images; these features can then be used for the classification of new faces (*Turk and Pentland, 1991*). In this procedure, each 'eigenface' (i.e. the eigenvectors resulting from PCA of all face images) represents a different dimension of facial variability and each face image can be represented by a series of weights associated with each eigenface. This creates a multi-dimensional 'face space' in which faces are represented as points based on their eigenface weights, and zero weights for all eigenfaces (i.e. the center of the space) represents the average face across all images. Such face spaces have psychophysical parallels in primate face processing systems (*Leopold et al., 2001*; *Cowen et al., 2014*; *Chang and Tsao, 2017*). Multiple images of each subject were averaged to generate average individual faces, which in turn were used to generate the average species faces that were used in eigenface decomposition. We classified new images using a nearest-neighbor classifier based on minimum Euclidean distance to each average species face in face space. This scheme corresponds to an average face model of guenon face learning, which assumes that guenons cognitively encode different species' face patterns as the mean of all encountered examples. An alternative exemplar model assumes that guenons encode example faces representing different species individually and compare new faces to the model. In previous work using similar methods, results generated based on average face and exemplar models produced very similar results (*Allen et al., 2014*), suggesting that these results are robust to the choice of learning model.

To avoid using the same individual guenons to both train and test our species classifier we used a leave-one-out procedure for all analyses. For this procedure, we systematically removed each individual from the image set, repeated the analysis procedure outlined above, then classified each image of the excluded individual based on the features generated from all other images. All species included in these analyses are represented by at least four individuals (range: 4–23). We present results for all species, however results for species with samples sizes in the lower end of this range should be considered less robust and interpreted with caution.

Eigenface-based features can be used to reliably classify guenons by species based on axes of variation, however the extent to which specific facial characteristics are relevant for correct

classification of each species is difficult to determine. We used an occlude-reclassify scheme developed to identify which image regions contribute most to correct classification in computer vision classification tasks (*Zeiler and Fergus, 2014*). For each correctly classified image, we systematically blocked each image region and re-classified the image; a correct re-classification indicates that the occluded region of the face was unnecessary for correct classification, while an incorrect re-classification indicates that the occluded region was essential. Occlusion of face regions was accomplished by setting the relevant pixel as well as all those in a thirty-pixel radius to the mean face color of that species. This procedure was repeated for every pixel in the image, effectively sliding the occluded region across all face areas. A radius of thirty pixels occludes approximately 5% of the image (*Figure 5*), with the specific region being occluded shifting by one pixel at each iteration. Primate faces are bilaterally symmetrical, therefore to avoid the presence of duplicate spatial information that may support species classification when part of the face is occluded, we ran analyses on the left and right halves of the face separately. Results differed little, so for clarity we report the results from the left hemi-face classification in the main text, with right-side results summarized in Additional Results in the Appendix. For more details on the implementation of the occlude-reclassify procedure, see Methodological Considerations in the Appendix. Based on this occlude-reclassify scheme, we generated a binary image for each image in our data set, with each pixel being either zero (black) or one (white) base on whether the image was correctly classified when that pixel and its neighbors was occluded. We then averaged these binary images across individuals and species to generate species

level heatmaps depicting face regions that are essential for correct classification across species. For visualization, we converted greyscale heatmaps to color using a color mapping function. To facilitate the identification of critical face regions, occlusion results are presented as composite images combining heatmaps and a greyscale version of the relevant species average face, with transparency set to 0.5.

Heatmaps vary across species in the extent to which face regions identified as essential for correct species classification are spread across the face (i.e. ranging from small and isolated face regions to large portions of the face identified as critical) as well as the relative import of identified regions (i.e. the likelihood that identified regions caused misclassification, encoded as how dark identified regions are in the heatmap). To quantify the spread and relative importance of the identified face regions across species, we calculated the proportion of the face misclassified and the mean classification error, respectively. The proportion of the face misclassified was calculated as the number of heatmap pixels less than one (i.e. those that were ever incorrectly classified) divided by the total number of pixels in the average face for each species; higher values indicate that the face regions essential for correct species classification are spread more widely across the face. The mean classification error was calculated as the mean value of all heatmap pixels less than one; higher values indicate that the face regions identified are particularly critical and more often lead to misclassification when occluded (i.e. the identified regions are darker in the heatmaps). Computational analyses were conducted in

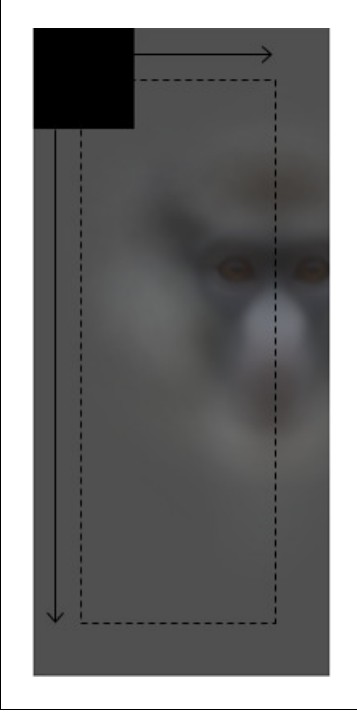

**Figure 5.** Average guenon face with an occluder shown in the top left. The occluder is depicted in black for maximal visibility, but in analyses presented here is set to the mean face color of the relevant species. During the occlude-reclassify analysis, the occluder is slid across the image and the image re-classified; an incorrect classification at a given occluder location indicates the presence of face information critical to correct classification. Analyses are run on hemi-faces to account for facial symmetry. Image borders outside the radius of the occluder are not tested; the dashed line encloses the region of the image analyzed using the occlude-reclassify procedure.

MATLAB and run on the High Performance Computing cluster at New York University.

The procedures described above were also used to run occlude-reclassify analyses using two subsets from the full image set. The first included putty nosed monkeys (*C. nictitans*) and all guenon species in the database that occur within their range: *A. nigroviridis*, *C. ascanius*, *C. diana*, *C. erythrotis*, *C. mona*, *C. neglectus*, *C. petaurista*, *C. sclateri*, *Ch. tantalus*, and *E. patas*. The second included mona monkeys (*C. mona*) and all guenon species in the database that occur within their range: *C. erythrotis*, *C. nictitans*, *C. petaurista*, *C. sclateri*, *Ch. tantalus*, and *E. patas*. These subset analyses represent a more biologically realistic discrimination test for the two guenon species for which we conducted experimental looking time trials. For both subsets, we included all species with geographic ranges that overlapped the range of the target species (*C. nictitans* or *C. mona*) by greater than 1%, and so these analyses were based on all taxa that each species could encounter in the wild. It is notable that in guenons, hybridization has been observed between species with differing habitat preferences (e.g. *C. mona* and *C. nectlectus*, *Chiarelli, 1961*; *C. mitis* and *Ch. pygerythrus*, *de Jong and Butynski, 2010*) and of differing body sizes (e.g. *Ch. sabaeus* and *E. patas*, *de Jong and Butynski, 2010*). Range maps for all species were obtained from the International Union for Conservation of Nature (IUCN) and range overlap was calculated as described previously (*Allen et al., 2014*).

## Looking time experiments

### Experimental population

Looking time experiments were conducted at CERCOPAN sanctuary in Calabar, Nigeria, and included 18 adult putty nosed monkeys (*C. nictitans*; n males = 6, n females = 12; mean (range) age = 13 (7-22) years) and 16 adult mona monkeys (*C. mona*; n males = 10, n females = 6; mean (range) age = 8.8 (3–24) years). All subjects received appropriate primate diets, environmental enrichment, and veterinary care, and were socially housed with other guenons in groups of two to five (mean = 4). All subjects were within visual range of heterospecific guenon species, which included *C. erythrotis*, *C. mona*, *C. nictitians*, *C. preussi*, *C. sclateri*, *Ch. tantalus*, and *E. patas*, although the exact heterospecific species visible varied by group. Primate species housed at CERCOPAN are all native to Nigeria. While some individuals were born in the wild, and others in captivity, we assumed that all individuals had potentially been exposed to all sympatric heterospecifics and treated each subject as if it were potentially familiar with both conspecific and sympatric heterospecific guenon faces.

Each species was divided into four experimental groups, which served as experimental replicates. Many experimental groups were also complete social groups, however in some cases social groups were combined to create experimental groups (e.g. two social groups containing only two mona monkeys were combined to yield an experimental group containing four subjects). Targeting specific subjects within a socially housed group is extremely difficult, therefore subjects within each experimental group received the same experimental treatments (i.e. they viewed the same stimulus image pairs in the same order), and trial characteristics were counterbalanced between experimental groups. Putty nosed monkey experimental groups contained either four or five subjects (mean = 4.5), and all mona monkey experimental groups contained four subjects. In each species, two experimental groups were presented with male stimulus images and two with female stimulus images across all trials.

### Stimulus image preparation

Experiments involved the simultaneous presentation of two stimulus images to subjects, with their resulting eye gaze measured to determine visual biases. Stimulus preparation and experimental procedures were carried out following the recommendations of *Winters et al. (2015)*. We obtained stimulus images from our image database of guenon faces, which were prepared in a similar way to those used for computational analyses. We cropped images to display the head and shoulders of the subject; applied color constancy using a combination of the gray world assumption and maximum point of reflectance, as described previously (*Allen et al., 2014*); standardized blur across images to a blur metric (*Crete et al., 2007*) of 0.3; and transformed images such that the outer corners of the eyes were level and 150 pixels apart. Standardized naturalistic backgrounds are preferable in looking time tasks (*Fleishman and Endler, 2000*), therefore we segmented the subject from

the background and replaced the latter with a blurred image depicting natural guenon forest habitat (taken in Gashaka Gumti National Park, Nigeria). We did not use a standardized reference frame (e.g. using the eyes as landmarks to standardize the positioning of the face) across all stimuli because this yielded images that were strangely framed due to the variability in facial characteristics and positioning (e.g. an excessive amount of background space on some sides of the face). Instead, we positioned the subject on the background image such that the framing appeared natural. This processing pipeline yielded a standardized set of stimulus images presented at approximately life-size. Stimulus image manipulations were conducted using the Image Processing Toolbox in MATLAB (*The MathWorks, Inc, 2019*) and the GNU Image Manipulation Program (*The GIMP Development Team, 2014*).

We matched each stimulus image pair for sex, with each subject viewing images of either all males or all females to facilitate comparisons of subjects across trials. We were unable to standardize eye gaze across all stimulus images, and so accounted for this factor statistically by incorporating stimulus eye contact as a variable in our analysis. All stimuli depicted mature adults without any type of aberrant facial appearance (e.g. scars, abnormal facial characteristics, hair loss, etc.). Stimulus images presented to mona monkeys were all from individuals outside CERCOPAN and were therefore all unfamiliar. CERCOPAN is the only source of putty nosed monkey face images in our database, therefore for their trials it was necessary to use images collected on site as stimuli. We minimized the familiarity between subjects and animals represented in stimulus images as much as possible; in some cases, it was necessary to present a stimulus image of an animal in visible range, however this was always at a distance and subjects never saw stimuli of animals in their own or adjacent cages. To account for any impact of individual familiarity we included this as a binary variable in statistical analyses of putty nosed monkey eye gaze (not within visible range vs. within visible range).

Our stimulus image set consisted of four images (two male, two female) of each heterospecific species: *C. ascanius*, *C. diana*, and *C. wolfi*; six images (three male, three female) of mona monkeys; and twelve images (six male, six female) of putty nosed monkeys. Mona monkey subjects were presented with all relevant stimulus images of one sex (each experimental group saw either male or female images). Putty nosed monkey subjects were presented with all relevant heterospecific stimulus images of one sex, but each experimental group was presented with unique putty nosed monkey stimulus images (sex-matched to heterospecifics) to reduce familiarity effects as much as possible.

## Experimental apparatus

We designed an experimental apparatus (*Figure 1—figure supplement 1*) that required guenons to view stimuli through a viewing window, effectively forcing them to remain relatively stationary while participating in experiments. Our experimental apparatus consisted of an opaque box (58 × 48×35 cm) housing all experimental equipment, with a small (14 × 6 cm) subject viewing window on one side. Inside the box we positioned a widescreen laptop (Acer ES1-711-P1UV) opposite the viewing window and a video camera (Canon Vixia HF R40 HD) centered immediately above the laptop screen to record the subjects' responses. The laptop LCD screen was color characterized weekly using ColorMunki Design (*Dubuc et al., 2016*), with the resulting color profile used to ensure the accurate presentation of colors in all stimulus images. To help interest subjects in the experimental apparatus, the external side of the apparatus that contained the viewing window was painted with a colorful pattern using watercolors. The pattern was varied across trials to facilitate sustained interest throughout the duration of experiments. Subjects could not see this colorful portion of the apparatus when looking through the viewing window, but we nonetheless included the external pattern used for the trial in statistical analyses to account for any impact this may have had on looking biases. The video camera records both video and sound, allowing researchers to dictate the names of subjects who look inside the apparatus to the camera. To minimize distractions, we typically said subject names aloud after subjects had moved away from the apparatus. Most subjects were inherently interested in the experimental apparatus, however in some cases food rewards were used to encourage participation (n = 5 trials involving n = 3 subjects). Food rewards included peanuts or diluted juice presented in a water bottle, selected based on subject preferences. The presence of food is unlikely to influence species discrimination capabilities or visual biases for faces.

## Experimental procedures

Our experimental trials involve the simultaneous presentation of paired con- and heterospecific faces, focusing on a particular facial trait for each species. For putty nosed monkeys we focus on nose spots and for mona monkeys on eyebrow patches, on the basis that each of these features is within the region of the face identified by our machine learning approach as being critical for that species. Each subject participated in three trials, with stimulus image pairs depicting the following: (1) a conspecific and a heterospecific that shares a focal trait with the conspecific, (2) a conspecific and a heterospecific that does not share a focal trait with the conspecific, and (3) a conspecific for which the focal trait has been modified and a heterospecific that shares the focal trait with the conspecific (for example stimuli, see *Figure 1*). Heterospecifics presented to putty nosed monkeys were Wolf's guenons (*C. wolfi*, no nose spot) and red-tailed monkeys (*C. ascanius*, nose spot); heterospecifics presented to mona monkeys were red-tailed monkeys (no eyebrow patches) and Diana monkeys (*C. diana*, eyebrow patches). Heterospecific species were selected based on the presence/absence of the relevant facial trait, a lack of range overlap with the subject species, and availability of sufficient and appropriate images in our database. We chose to focus on allopatric heterospecifics because these tend to share more facial traits with the tested species (e.g. no locally endemic species have a white nose spot similar to that of a putty nosed monkey). Image presentation locations (i.e. left verses right) and trial order were counterbalanced across experimental conditions; both factors were included in statistical analyses. For each trial, we placed the experimental apparatus immediately outside the relevant enclosure and recorded the identities of participating subjects. We waited a minimum of one week between trials of the same subject to minimize habituation or trial order effects.

## Video coding

Videos of each trial were coded frame by frame to quantify the amount of time subjects spent looking at each stimulus image. All coding was done blind to trial conditions and stimulus image location. Reliability was assessed using approximately 10% of all trial videos, in which we assessed agreement between two coders on the direction of jointly coded looks within these trials as being in agreement in 94.46% of frames (Cohen's kappa = 0.883), which is well within the range of acceptable reliability scores for this type of data (*Winters et al., 2015*; *Dubuc et al., 2016*). Raw looking time data was compiled to yield a total number of frames spent looking at each stimulus image for each subject in each trial. Subjects varied widely in their level of interest in experiments, resulting in considerable variation in overall looking time. To generate a dataset with relatively similar overall looking times we used only the first five seconds of looking for each subject in each trial, while allowing them to complete the current look at the five seconds mark (i.e. we required at least one second of non-looking before terminating coding for each subject). This threshold was chosen because the majority of subjects viewed images for less than five seconds, and because it yielded durations similar to those reported in previous looking time experiments in primates (*Higham et al., 2011*; *Winters et al., 2015*; *Dubuc et al., 2016*). This resulted in a mean total looking time (± standard deviation) of 3.89 s (± 1.98 s) for putty nosed monkeys and 4.58 s (± 2.52 s) for mona monkeys. Because a direct comparison is made between the species depicted in stimuli, each trial effectively serves as its own control.

## Statistical analyses

We used R to analyze differences in looking time elicited by subjects in experimental trials using generalized linear mixed models (GLMMs). Models were fit using a binomial family distribution, with the number of video frames spent looking at the targeted stimulus image and the number of video frames spent looking at the paired image set as the binomial outcome variable, bound together in R using the cbind function (*Zuur, 2009*). We confirmed that the residual variance of all models conformed to a binomial distribution, and that this variance structure was therefore appropriate, by inspecting Q-Q plots (generated using R package 'car' version 3.0–2, *Fox and Weisberg, 2011*) of model residuals versus binomial quantiles generated based on a binomial distribution with the size parameter set to the mean number of total looking frames across subjects (putty nosed monkeys: 117, mona monkeys: 137) and the probability parameter set to 0.5. This modeling structure analyzing paired counts of looks at simultaneously presented stimuli allowed us to assess looking biases

while accounting for any differences in total looking time across subjects. All models included group, subject, and unique trial (i.e. a unique identifier for each subject in each trial, included to account for our analysis of the two images presented in each trial as separate data 'rows') as nested random effects. Stimulus species (conspecific v. heterospecific) and focal trait similarity (presence of nose spots for putty nosed monkeys and eyebrow patches for mona monkeys), were included as fixed effects. We also included the following additional factors as fixed effects: subject age (log transformed), sex, and origin (captive v. wild born); stimulus image presentation spot (right v. left), eye contact (direct eye contact with the camera v. looking slightly away), sex, and degree of familiarity to the subject; and trial order, apparatus pattern, and display ICC profile.

To determine which variables significantly influenced subject looking biases, we compared models with different parameterizations using likelihood ratio tests (LRTs). A single model including all fixed effects simultaneously would involve an excessive number of predictors. We therefore first analyzed each variable separately via comparisons to a null model including only random effects, and excluded non-significant predictors from subsequent analyses. We generated an initial model composed of factors that were statistically significant (alpha <0.05) or exhibited a trend (alpha <0.1) when tested alone. To determine the statistical significance of these factors, we then systematically excluded each factor from this model and tested its contribution to the fit of the model to the data using LRTs. When species (conspecific v. heterospecific) and focal trait (shared v. not shared) were both significant predictors in this model we also tested a species*trait interaction. Within a final model composed of significant predictors we compared across factor levels of fixed effects using z scores calculated using a normal approximation. We report conditional odds ratios as measures of effect size for statistically significant predictors (*Gelman and Hill, 2007*; *Hosmer et al., 2013*). Adherence to model assumptions was verified based on plots of fitted values and residuals. Trials from putty nosed and mona monkeys were analyzed separately. For each species we ran two models: one modeling significant predictors using data from trials including a conspecific and a heterospecific without the species-typical face trait, which evaluates visual biases when there was no conflation between species and species-typical traits (experimental condition one); and a second using data from all experimental conditions to evaluate how the availability of species-typical information influences eye gaze. GLMMs were run using the 'lme4' package version 1.1-18-1 (*Bates et al., 2015*) in R version 3.5.2 (*R Development Core Team, 2018*).

## Acknowledgements

We thank Allegra Depasquale and Laura Newman for assistance coding looking time videos, and Kathryn Yee for assistance with stimuli preparation. Special thanks to the directors and staff of CERCOPAN sanctuary, particularly Claire Coulson and Isabelle Theyse, for access to the facility and support during data collection. We also thank the editor, PJ Perry, and Tim Caro and two anonymous reviewers for constructive and highly useful feedback on the manuscript.

## Additional information

### Funding

| Funder | Grant reference number | Author |
| --- | --- | --- |
| American Society of Primatologists | General Small Grant | Sandra Winters |
| National Science Foundation | Doctoral Dissertation Research Improvement Grant 1613378 | Sandra Winters James Higham |
| National Science Foundation | IGERT Fellowship | Sandra Winters |
| New York University | MacCracken Fellowship | Sandra Winters |

The funders had no role in study design, data collection and interpretation, or the decision to submit the work for publication.

## Author contributions
Sandra Winters, Conceptualization, Data curation, Software, Formal analysis, Funding acquisition, Validation, Investigation, Visualization, Methodology, Project administration; William L Allen, Resources, Methodology; James P Higham, Conceptualization, Resources, Supervision, Funding acquisition, Methodology, Project administration

## Author ORCIDs
Sandra Winters (iD) https://orcid.org/0000-0002-9561-5950
William L Allen (iD) https://orcid.org/0000-0003-2654-0438
James P Higham (iD) https://orcid.org/0000-0002-1133-2030

## Ethics
Animal experimentation: This study was performed in strict accordance with the recommendations in the Guide for the Care and Use of Laboratory Animals of the National Institutes of Health. All experiments were conducted according to institutional animal care and use committee (IACUC) protocols (#15-1466) approved by the University Animal Welfare Committee of New York University.

## Decision letter and Author response
Decision letter https://doi.org/10.7554/eLife.47428.sa1
Author response https://doi.org/10.7554/eLife.47428.sa2

# Additional files

## Supplementary files
• Supplementary file 1. Occlude-reclassify machine classification results.
• Supplementary file 2. Looking time results.
• Transparent reporting form

## Data availability
All project data and code are available in Zenodo repositories under the following DOIs. The guenon face image database is available in repository https://doi.org/10.5281/zenodo.3572780, and code implementing the occlude-reclassify analysis is available in repository https://doi.org/10.5281/zenodo.3574512. Experimental looking time data and original videos are available in repositories https://doi.org/10.5281/zenodo.3572791 (Cercopithecus mona) and https://doi.org/10.5281/zenodo.3572789 (Cercopithecus nictitans), and code implementing looking time analyses is available in repository https://doi.org/10.5281/zenodo.3574529.

The following datasets were generated:

| Author(s) | Year | Dataset title | Dataset URL | Database and Identifier |
|---|---|---|---|---|
| Sandra Winters, William L. Allen, James P. Higham | 2019 | The structure of species discrimination signals across a primate radiation: guenon image database | https://doi.org/10.5281/zenodo.3572780 | Zenodo, 10.5281/zenodo.3572780 |
| Sandra Winters, William L. Allen, James P. Higham | 2019 | The structure of species discrimination signals across a primate radiation: Cercopithecus mona looking time trials | https://doi.org/10.5281/zenodo.3572791 | Zenodo, 10.5281/zenodo.3572791 |
| Sandra Winters, William L. Allen, James P. Higham | 2019 | The structure of species discrimination signals across a primate radiation: Cercopithecus nictitans looking time trials | https://doi.org/10.5281/zenodo.3572789 | Zenodo, 10.5281/zenodo.3572789 |

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

Appendix 1

## Occlude-reclassify procedure

### Methodological considerations

In this study, we used an occlude-reclassify procedure to identify face regions critical for correct machine classification of guenon faces by species. The general approach of this procedure is straightforward: different signal components (i.e. face regions in this study) are systematically occluded and the image reclassified; essential signal components are those that cause mis-classification when occluded. However, several details must be considered when implementing this procedure using images. We overview some of these here.

### Occluder color

The occluder that masks each image region must generally be set to a color, and there are multiple options for this. The goal of the procedure is to negate the given region, and the best method of doing so will differ across circumstances. In some cases, it may be possible to model the occluded region as missing data (e.g. as NaN in MATLAB), however this can introduce computational problems (e.g. the PCA-based method used here requires complete data) or complicate interpretations (e.g. there are no missing data in natural scenes, making this less biologically realistic). Here, we employed two different occluder colors: (1) the average gray of all guenon faces (i.e. the color used for the background of all images), and (2) the mean face color (calculated without including the background) of the species in question. Average gray occluders have been used previously (*Zeiler and Fergus, 2014*), are convenient to implement, and generate results that are directly comparable across groups. However, using the same color occluder for all groups potentially biases classification performance as classification should be better for species that have more average color faces than species with very light or dark faces. Furthermore, in this project we compared computational results derived using the occlude-reclassify procedure to results from experiments in which facial stimuli were presented to guenon subjects. In those experiments some facial stimuli were modified, with certain face regions (either nose spots or eyebrow patches) altered to better match the rest of the face. A species-specific occluder color is more in line with these experimental manipulations and we therefore present these results in the manuscript. In setting the color of the occluder to the species-specific mean face color we introduce a different potential bias, effectively increasing the likelihood of detecting face regions that are salient against the general face color of the species. However, these regions will also be more salient to receivers, especially since our analyses are conducted in guenon color space. Given that our goal is to identify face regions most likely to contribute to species discrimination, this bias seems reasonable. For completeness, we have also included results generated using a standard average gray occluder (see *Figure 2—figure supplements 1–11*).

### Occluder size

The size of the occluder must also be specified. This choice reflects a trade-off: for very small occluders, misclassification will be very rare and essential signal components will not be identified; for very large occluders, misclassification will be very common and the identified signal components will converge on the entire area of the signal. We chose an occluder size that balanced these two extremes, being large enough to detect face regions of interest, but small enough that the regions identified were restricted to only some regions of the face and therefore useful for identifying critical facial features.

## Occluder shape

Occluders can also vary in shape. In our analyses we used a square occluder, which will often be the most reasonable choice. Other shapes may sometimes be more useful, however, depending on the characteristics of the signal of interest. For instance, long rectangular occluders may be useful for analyzing stripes, or more complex patterned occluders (e.g. multiple spots) for analyzing signals with repeated pattern elements.

A potential alternative use of the occlude-reclassify procedure is to test multiple occluders of different shapes to determine which causes the most disruption of correct classification and therefore represents the most biologically meaningful pattern.

## Signal symmetry

Some signals, such as the faces analyzed here, are symmetrical. Sliding an occluder across an entire image that depicts a symmetrical signal may be problematic because mirrored features will never be fully occluded. For instance, if cheek tufts in one species and a nose spot in another species are equally critical for correct classification, cheek tufts – which exist in pairs on opposite sides of the face – may nonetheless be less likely to be identified using the occlude-reclassify procedure because when one is occluded the other is still visible to the classifier. By comparison, a single nose spot in the middle of the face could be occluded in its entirety, completely masking this information and potentially making this feature more likely to be identified as critical using this scheme. We address this by classifying hemi-faces, in which the signal is split along the line of approximate symmetry. In our implementation of the occlude-reclassify procedure the occluder location is recorded based on the central pixel, so there is a buffer along the edges of the images (equal to one half of the occluder length) in which critical signal regions are not detected. The edges of the full images are simply gray background, so this is not problematic, however when generating hemi-face images a buffer region must be added alongside the midline of the face to allow the occluder to reach the center of the face. In order to give midline regions of the face equal importance in the classification we used the relevant portion of the other half of the face as a buffer. Prior to analysis we arbitrarily chose to present results based on the left hemi-face (from the perspective of the viewer; the right side of the animal's face). These are similar to results based on the left hemi-face, although not identical (see *Figure 2—figure supplements 1–11*).

## Additional Results

For completeness, we ran our occlude-reclassify procedure using two occluder colors and both left and right hemifaces. We present results generated using species-specific mean face colored occluders and left hemifaces in the Results section of the manuscript; here, we present and discuss the remaining results.

We used two occluder colors to satisfy different conditions: a species-specific mean face color occluder is consistent with the goal of the study to identify how discrimination is supported by signal traits in different species, however using the same gray occluder across species facilitates comparisons across species. Results based on the species-specific mean color (left hemi-faces: *Figure 2*; right hemi-faces: *Figure 2—figure supplement 9*) and the average gray (left hemi-faces: *Figure 2—figure supplement 10*; right hemi-faces: *Figure 2—figure supplement 1*) are very similar for some species. For instance, the results for *C. ascanius* and *C. lhoesti* are virtually identical using the two methods; this is unsurprising because the mean face color for each of these species is very similar to the overall average face color used for the gray occluder. However, for species with generally darker or lighter faces, results differ between the two methods. For instance, *M. talapoin* and *C. nictitans* are among the lightest and darkest faced guenon species, respectively, consequently results differ depending on occlude color. Again, this is unsurprising because the occluder itself is salient against these faces, rather than muting a face region as in other species. In these cases the gray occluder is less realistic, thus our choice to present the species average face color results in the main section of this manuscript. It is worth noting that our experimental results

documenting putty nose monkey (*C. nictitans*) visual biases are much more strongly paralleled by occlude-reclassify results based on a species-specific mean face color occluder. This is particularly relevant because the modifications made to putty nosed monkey stimulus images used in experiments involved darkening the white nose spot to match the rest of the face, a procedure that corresponds to using a darker occluder.

We also ran occlude-reclassify analyses using right and left hemi-faces separately, to account for any effects of bilateral symmetry in guenon faces. Across species there are broad similarities in the parts of the face identified as critical for correct species classification using the left (species-mean occluder: *Figure 2*; gray occluder: *Figure 2—figure supplement 10*) and right hemi-faces (species-mean occluder: *Figure 2—figure supplement 9*; gray occluder: *Figure 2—figure supplement 1*). In some cases a feature is identified as having greater relative importance (i.e. a higher rate of misclassification) on one side compared to the other (e.g. the eyebrow region of *C. wolfi*), and in others a feature is only identified on one side (e.g. the cheek tufts of *Ch. sabaeus*). Differences are likely due to asymmetries in posture introduced because of the need to photograph unrestrained and mobile animals. More pronounced differences are found in some species with low sample sizes, reinforcing that results for these species should be interpreted with caution. Despite some variation in results from each side of the face, the general patterns observed in our occlude-reclassify results are similar and our conclusions remain unchanged. Importantly, results from both putty nosed monkeys (*C. nictitans*) and mona monkeys (*C. mona*) are very similar for the two hemi-faces, with our classifier relying exclusively on the nose spot (albeit with a stronger effect observed in the right hemi-face) and across broad face regions, respectively. Therefore, variation between results generated based on different hemi-faces does not influence our interpretations of the parallels between computational and experimental results in these two species.

