## [Decision Letter]

**Acceptance summary:**

This study by Winters et al. makes significant contributions to our understanding of species discrimination processes by combining machine learning-based classification of facial feature patterns with a clever behavioral experiment approach in guenons, a closely-related group of primate species who often inhabit the same forests. In such situations, there may be strong selection on signals or cues (which could be phenotypic, olfactory, vocal, behavioral, or combinations thereof) to help individuals reliably identify other members of their same versus different species, for example to help them avoid mating with individuals from different species. Guenons have an incredible diversity of colorful facial feature variation. In their study, Winters et al. first used a computational approach to predict which facial regions might be most critical for distinguishing each species from others. They then performed behavioral "looking-time" experiments with individuals from two guenon species. One of these species has a primary defining facial feature characteristic that was predicted to be highly discriminatory on its own; in contrast, discrimination for the other species is more dependent on the combined presence of multiple facial coloration traits. The investigators presented individual monkeys with pairs of images, both representing actual coloration patterns of the same and different species (with individuals gazing significantly longer at images representing their own species) and with manipulations to represent artificial combinations of the presence or absence of the respective most discriminating feature. Individuals from the species with the dominant discriminatory feature gazed longer at images with that feature present, yet the same result was not observed among individuals from the species for which the top-ranked feature was predicted to be much less discriminatory on its own. While the looking-time results cannot be directly linked to mate choice behavior, the elegant design of this integrative study and its clear results provide insight into the diverse and multidimensional mechanisms that could plausibly be involved in such processes, even within a closely-related group of species.

**Decision letter after peer review:**

Thank you for submitting your article "The structure of species discrimination signals across a primate radiation" for consideration by *eLife*. Your article has been reviewed by four peer reviewers, including George H Perry as the Reviewing Editor and Reviewer #1, and the evaluation has been overseen by Andrew King as the Senior Editor. The following individual involved in review of your submission has agreed to reveal their identity: Tim Caro (Reviewer #4).

The reviewers have discussed the reviews with one another and the Reviewing Editor has drafted this decision to help you prepare a revised submission.

Summary:

This paper uses an integrative machine learning and behavioral experiment approach to examine facial coloration pattern variation among a radiation of geunon monkeys with respect to potential species recognition mechanisms. For two species, facial features classified as most important for identification of that species relative to other guenons were altered in photos, which were then displayed to test subjects of each species in a 3-part paired design (involving unaltered conspecific and heterospecific (both for species sharing and not sharing the dominant classification feature with the test species) and manipulated conspecific images) in a looking-time experiment. The experimental results differ between the two test subjects in a manner consistent with that which would have been predicted by the machine learning classifier in terms of the discriminatory power of the dominant feature versus necessity of multiple features in combination.

We had an extensive and productive reviewer consultation session about your manuscript. As the reviewing editor I ultimately chose to recommend a 'needs revisions' decision for your article because first, the reviewers believe that our essential revisions (listed below) can be conducted within a two month timeframe, and second, we would be enthusiastic about the article if (following re-evaluation) the revised analyses prove robust and if the biological significance of the results could be convincingly clarified. Yet I would like to emphasize that the reviewers have high expectations on these points; given their fundamental nature the revised manuscript would be thoroughly re-assessed. Given this status, I am also choosing to break the typical *eLife* mould by including the full reviews below, because they provide helpful context for the summarized essential revisions, while also making multiple other excellent points that I believe you would want to consider and address accordingly so that your work can reach its top potential.

Essential revisions:

1) Demonstrate that similar classifier results are obtained within a more realistic taxonomic context. That is, for each of the two test species, the model should ideally consider the species recognition problem at hand, e.g. with sympatric taxa only. If impossible given image data availability, at the least the authors should show that results with different subsets of available taxa are similar.

2) Analyze data from the three different types of trials separately; the grouping of data from across the different trial types could conflate results. Additionally, correct the specific statistical analysis approach errors as detailed by Reviewer #3.

3) Clarify the biological significance of the looking-time experimental results. The relationship of these results with species recognition processes was considered indirect at best and inferential. Specify what can concluded once these results are clarified; is this still an impactful result? If so, emphasize that aspect, while also still discussing potential implications for species recognition. As the reviewing editor, I noticed that when you had more space to describe your results you were appropriately careful (yet this was not the case in several summary sentences). However, even then I think it would be better to be more direct about this limitation, while simultaneously celebrating the explicit knowledge that is obtained if warranted.

Reviewer #1:

Overall I am highly impressed by this study and its contributions to our understanding of visual species recognition systems. The novel combination of computational prediction with experimentation in two non-human primate species is outstanding. The contrasting results between putty-nosed and mona monkeys are compelling and informative.

Specifically, individuals from both species gazed significantly longer at unaltered images of conspecifics versus heterospecifics when shown side-by-side (both for when the heterospecific species shown shared the predominant classification feature with the test species and when it did not). All monkeys were also exposed to an experimentally manipulated image of a conspecific species but for which only the predominant classification feature was switched to that from a dissimilar species side-by-side with an unaltered image of a heterospecific species but that shares the predominant classification feature with the test species.

In putty-nosed monkeys, the predicted highly discriminant nose-spot feature significantly explained gazing time, even in the manipulated conspecific vs. unmanipulated heterospecific experiment. The opposite was true for mona monkeys (white eyebrow patch feature), for whom the classifier had predicted a lower level of reliance on the predominant feature compared to putty-nosed monkeys. The study design used (including various condition randomizations) and statistical analyses were appropriate.

This study represents an outstanding experimental confirmation of an automated visual species recognition prediction for a species with a highly discriminant feature (putty-nosed monkeys), along with the mona monkey result in contrast demonstrating the combinatorial importance of multiple facial features (including when the otherwise top-ranked discriminant feature for that species is manipulated artificially!), highlighting species recognition system variability.

Essential revisions:

1) The authors should run the manipulated images that were used for their experiments back through the classifier (i.e. the approach used to generate the images for the experiments is different than the blurring approach used in the machine learning process) for confirmation that the level of uncertainty introduced matched the experimental results.

2) I was surprised that some key information on the monkeys included in the experiment was only available in the supplementary information. I had noted a number of larger concerns while reading the paper (e.g. potential lack of history of exposure to heterospecifics) that were then mostly alleviated when I explored the supplementary information for answers, and I expect the experience might be the same for future readers. Therefore I would encourage this information to be brought into the main manuscript Introduction/Results and Materials and methods.

2B) Were any of the sanctuary monkeys born in the wild (and thus potentially experienced heterospecific interactions outside of the caged sanctuary environment?). If so and if this information is known, is there a significant difference with captive-born individuals? Should this variable be included in the analytical model?

Reviewer #2:

This study addresses the role of facial features in species recognition in guenons. It includes two parts. In the first part, the authors used a computer vision technique (machine classification with partial occlusion applied onto face images encoded by eigenface scores) to identify which facial features influence species classification most. In the second part, they designed a behavioural experiment with two guenon species to test whether key features identified in the first part influence looking time in conspecifics.

This is the first study that uses classification+occlusion to identify potentially relevant features. This is a great methodological step forward in visual ecology. However, I feel that neither the classification part nor the behavioural experiments are appropriately designed to test the proposed hypotheses, and thus I am not sure that results do support conclusions. More specifically, the author's central statement that their "results objectively identify the signal components most likely to be useful to guenon receivers" may be incorrect, for two main reasons.

First, their machine classification, as any artificial or biological classification process, is not objective. Any classification or recognition task is inherently contextual. If a classifier, be it artificial or biological, was instructed to recognize red Ferrari among yellow Lamborghini and blue Hummer, it would not rely on the same cues as if it was instructed to recognize the same Ferrari among red Lamborghini and red Hummer. Thus, which features are keys for recognition critically depends on which stimuli need to be classified. In this manuscript, I am puzzled by the selection of species included in the classification analysis. For example, it does not seem relevant to include, in the same classification task, Miopithecus, Erythrocebus patas, Cercopithecus ascanius and C. neglectus. Males of Miopithecus weight 1.3 kgs, while patas males weight around 12 kgs. Moreover, these two species live in large monospecific groups, not in polyspecific groups that include Cercopithecus spp.. Cercopithecus ascanius is arboreal, while neglectus is semi-terestrial. Actually, mixed-species groups concern mostly arboreal species; with groups often made of one species of each Cercopithecus 'super-species' complex cephus, ascanius and nictitans (sometimes intermittently further including colobus mangabeys and mangabeys). If the authors want to investigate which features matter for species recognition, they first need to clearly define the perceptual task/challenge that each species faces in the wild, thus considering known species associations, the ecology of species and a minimum of morphological data (to avoid pooling together species which weigh differs by factor 10!). The 2011 article by Jaffe and Isbell, cited in the manuscript, and some older contributions by field primatologists Annie Gautier-Hion, Marc Colyn and others would be very useful. If the authors do not want to do such an analysis, at least they should demonstrate that results (their identification of key features) are robust to the selection of species included in the classifier. But putting all guenon species into a single classifier is for sure biologically irrelevant, and so appear the results.

Second, and more importantly, it is unclear to me how the behavioural experiment can support any conclusion about species recognition. In humans, object or face recognition is achieved during the first few hundred milliseconds (e.g., Cichy, Pantazis and Oliva, 2014. Resolving human object recognition in space and time. Nature neuroscience / Hsiao and Cottrell (2008). Two fixations suffice in face recognition. Psychological science). Most likely other old-world primates also achieve conspecific recognition within a few milliseconds. Eventually, how a difference of visual attention measured during a viewing period lasting several seconds is related to recognition? Results of this study show that the white nose of *C. nictitans* caries information that is relevant for this species specifically or, as the authors wrote, "that longer looking time at a particular face reflects level of interest". But this study does not bring any evidence that this information is about species identity. The authors cite Mendelson and Shaw's review, which argues that species recognition traits are not different from other sexual signals. This is may true in some model species, yet, 1) it may be beneficial to advertise conspecific identities for interactions that are not sexual; 2) at least in one old world primate, *Homo sapiens*, it has been shown that face detection, recognition and evaluations correspond to different perceptual domains, i.e. they have different neural bases (e.g. Tsao and Livingstone, 2009, Mechanisms of face perception. Annual Review of Neuroscience / Zhen, Fang and Liu, 2013. The hierarchical brain network for face recognition. PloS one, 8(3), e59886.); 3) these different neural bases likely translate into different selective forces on signals and, accordingly, recognition traits undergo displacement, while all sexual signals do not. Recognition traits are special, distinct from classical sexual signals, and evidence that a trait is sexually or socially important is no evidence that this trait is relevant for recognition.

Moreover, a match between results from the machine classification and the behavioural experiment does not prove the relevance of the behavioural experiment for identifying recognition traits. I am pretty certain that a saliency analysis (e.g. using saliency maps) would have revealed that the nictitans' white nose and mona's eyebrow patches are salient. Saliency may be a property of discriminative traits (thought not necessarily, especially in species that use long-distance species-specific signals to communicate, like calls – see the various studies by Jean-Pierre Gautier on the calls of guenons and their use in mixed-specie groups), but it can also be a property of sexual signals that are not recognition traits (ie not under selection for displacement), of traits signalling kinship, individual identity etc… (though, again, even this result would barely support any conclusion, since saliency influences bottom-up selective attention, which cannot be investigated by measuring looking times lasting 3 or 4 seconds).

Reviewer #3:

With great interest I have reviewed the manuscript titled "The structure of species discrimination signals across a primate radiation". Here the authors present a clever approach, not only identifying unique species-specific features via machine-learning, but also determining whether these features are perceived by two of the species within the radiation. The question of how signals may guide character displacement and species recognition is fundamentally important to our understanding of behavior and evolution. Tackling this question with sophisticated analytical tools and behavioral tests (without operant conditioning) in monkeys while acquiring appropriate sample sizes is impressive. And importantly, the manuscript is very well written and provides a thorough background of the literature/theory and discussion of the results.

The authors interpret that longer looking times for a particular face as increased interest, which seems justifiable. And the three experimental trials in each species also provide an adequate test of the effects of the selected features on attention. I do wish there were additional experimental for a negative control, such that other areas of the face were manipulated to see if attention was affected. However, given the limitations of the study system, I completely understand.

I do have some concerns about how the data were analyzed and presented:

It is unclear to me why a binomial family distribution was used when the "number of video frames" is the dependent variable. Binomial models assume a binary response variable (e.g. 0 or 1). Although a binary choice is provided here for each trial, the data analyzed is count data (duration). Thus, a negative binomial distribution (or Poisson) would be appropriate. It appears to me that this could dramatically impact the statistical interpretation. Furthermore, when testing out the code and data reported here, the models report a "Singular fit", which indicates that the random effects may be too complex for the data provided.

Furthermore, I do not understand the choice to analyze all three trials data for each species together given that stimuli were presented in pairs. It is my understanding that if a subject is looking at 1 image during the five seconds window it cannot be looking at the other image – thus making each data point non-independent (and only relevant to the trial). Wouldn't it be easier for interpretation to just analyze each trial separately? For example, the Results section describes differences in con- vs hetero-specific looking times; yet only in trial 1 was this comparison between natural images of each species (e.g. I wouldn't predict a difference in con vs het in Trial 2).

Reviewer #4:

This is a very exciting paper that provides convincing evidence that guenon faces are a way to avoid hybridization. It combined careful analyses of faces with very interesting behavioral titration of attention by 2 of the guenon species.

---

## [Author Response]

Essential revisions:1) Demonstrate that similar classifier results are obtained within a more realistic taxonomic context. That is, for each of the two test species, the model should ideally consider the species recognition problem at hand, e.g. with sympatric taxa only. If impossible given image data availability, at the least the authors should show that results with different subsets of available taxa are similar.

As suggested, we have run additional analyses with two species subsets: those that overlap in range with putty nosed monkeys, and those that overlap with mona monkeys. The results from these two new analyses show substantial similarity with the original results based on the full image dataset. We now describe the analysis of these subgroups in the manuscript (Introduction fifth paragraph; Materials and methods subsection “Occlude-reclassify procedure”), and mention in the Results section that these are consistent with the general pattern (second paragaph). For the figure, we feel that the generality of the original results is preferable for presentation, but have now included depictions of our results utilizing putty nosed and mona monkey subsets in Figure 3—figure supplements 4-1.

2) Analyze data from the three different types of trials separately; the grouping of data from across the different trial types could conflate results. Additionally, correct the specific statistical analysis approach errors as detailed by Reviewer #3.

Regarding the analysis of trials, our experiments are designed to answer two key questions: (1) do guenons exhibit a conspecific bias, as do other primates? and; (2) how do manipulations of specific face traits influence this bias? These are the critical comparisons for our research questions, and so we structured the presentation and analysis as such. To answer the first question we analyze the trial comparing unmanipulated images of con- and hetero- specifics separately for both species. The results of this validate our approach, and show that, like other primate species, these two guenon species show a visual bias towards conspecific faces compared to heterospecific faces. Following this, to answer question 2, we analyze data from all experimental conditions to test how variation in the available face pattern stimuli in images affects visual attention. A unified analysis approach is preferable for this because we are more interested in overall drivers of visual attention than in the specific results of a given condition. We agree with the reviewers that the rationale for our analysis structure was unclear, and we have made changes to the manuscript to clarify this in several places: We have clarified our experimental conditions and now state that: “[The first] condition represents a baseline in which there is no conflation between species and the species-typical face traits, and confirms that the general pattern of visual biases for conspecific faces holds in the two guenon species tested here…[The second and third trials] ask the question: how is a conspecific bias affected by manipulations of which facial pattern components are available?…We present results analyzing visual biases in the first condition, as well as a unified analysis of visual biases across all experiments to identify the overall drivers of looking time across condition”.

Regarding the use of the binomial versus the Poisson distribution, in response to the reviewers concerns we have undertaken extensive and thorough revision of the use of different modelling distributions and we feel confident that we are correct to use a binomial distribution. Our analysis is structured in such a way that the dependent variable consists of pairs of counts in the form of n video frames looking at one image vs n video frames looking at the paired image. We explain here why Reviewer 3 is incorrect to say that “Binomial models assume a binary response variable (e.g. 0 or 1)”. The binomial distribution denotes the number of ‘successes’ out of n trials based on a given probability of success, and binomial models are run with paired count data denoting the number of ‘successes’ and ‘failures’ (or number of video frames looking at conspecifics v. number looking at heterospecifics). The Bernoulli distribution is a special case of the binomial distribution in which the number of trials is one, and therefore the response variable is indeed simply zero or one. Thus, while binomial models are sometimes used when a response variable is binary, they do not assume that the response variable is binary. Nor do they explicitly assume that the response variable is binomially distributed – only that the residual variance of the model is (though the residual distribution and response variable distribution often show structure of similar distribution). In the present case, we have a response variable that is binomially structured, and most importantly, that has a binomial error structure. We therefore feel confident that a binomial model is the most appropriate. This is supported by descriptions of how to analyze data structured as is our response variable, which consists of paired count data – see Zuur et al., 2009 "Mixed Effects Models and Extensions in Ecology with R" (binomial/Bernoulli distributions are described on pages 202-204, and the "GLM for Proportional Data" example that begins on p. 254 is relevant here). In addition, a number of relevant examples posted on statistical exchanges have similar data structures, and used the same approach as we did to analyze their data, supported by comments from senior statistical experts such as Ben Bolker.

https://stats.stackexchange.com/questions/189115/fitting-a-binomial-glmm-glmer-to-a-response-variable-that-is-a-proportion-or-f

https://stats.stackexchange.com/questions/89521/binomial-count-data-use-glmer-lmer-or-just-average-it-all

https://stackoverflow.com/questions/21441817/glmer-predict-with-binomial-data-cbind-count-data

https://stats.stackexchange.com/questions/230634/glmer-vs-lmer-what-is-best-for-a-binomial-outcome

We feel that the reviewer’s comments may have arisen because our description of the data structure and statistical methods was not as clear as it could have been. We have made revisions to the manuscript to make the response variable structure clearer, and to clarify that the use of a binomial error distribution is supported by the residual variance structure of our models (“Statistical analyses”).

Reviewer 3 noted that in their own reanalysis of our data “the models report a "Singular fit", which indicates that the random effects may be too complex for the data provided”. Our analyses using the version of R package lme4 reported in the manuscript (1.1-18-1) did not generate this warning, which was added in package update 1.1-20. (We note that the message is a warning message, not an error message.) Since the time of the review, the warning message text provided by lme4 (version 1.1-21) has been updated again to “boundary (singular) fit” and the authors of the package have provided discussion of this warning message here: https://github.com/lme4/lme4/issues/502. Briefly, the warning is generated when some linear combination of variance effects terms is zero. After investigating this issue, we are confident that this warning is not of concern for our data and conclusions. The boundary fit warning message is generated even when an estimate of zero is correct (i.e., groups do not vary more than expected by chance). For instance, the following R code using simulated data with randomly assigned groups (i.e., the variance of the random effect should be estimated at zero) generates the same boundary fit warning:

x <- runif(10000) #random samples from a uniform distribution

y <- rbinom(10000,100,0.5) #number of “successes” out of 100 trials when the probability of success is 50%

f <- sample(10,10000,replace=T) #randomly assigned groups

m <- glmer(cbind(y,100-y)~x+(1|f),family='binomial')

We followed a number of steps to verify that this new warning message, while helpfully flagging a potential issue, does not represent a real issue for our data analysis, including dropping random effects and running analyses in a Bayesian framework which includes priors for the covariance of the random effects and modeled coefficients. When we ran models that included simplified versions of our random effects structure (i.e. all combinations of one, two, or all three random effects), these models also returned the boundary fit warning. Because this analysis did not eliminate the warning message and the importance of including random effects that account for the nested structure of our data in analyses, we do not report these reduced models in our revised manuscript. Bayesian parameter estimation can avoid boundary fits because including priors for covariance structures prevents linear combinations of effects from being zero. We then re-ran our analyses in a Bayesian framework with Wishart priors, using the function bglmer in R package blme (Chung et al., 2013 Psychometrica, “A nondegenerate penalized likelihood estimator for variance parameters in multilevel models”). This generated results that were extremely similar to those returned using the glmer function in lme4, with all statistical tests generating the same qualitative result (final models for condition 1: putty nosed monkeys exhibited a conspecific bias (Chisq = 63.310, p < 0.001) and a right side bias (Chisq = 35.523, p < 0.001), mona monkeys exhibited a conspecific bias (Chisq = 30.756, p < 0.001); final models for analysis across all conditions: putty nosed monkeys exhibited a bias for nose spots (Chisq = 11.177, p < 0.001) and a right side bias (Chisq = 18.405, p < 0.001), mona monkeys exhibited a conspecific bias (Chisq = 177.477, p < 0.001), a bias for faces without eyebrow patches (Chisq = 29.456, p < 0.001), and a significant species*trait interaction (Chisq = 8.213, p = 0.004)). While we could report the Bayesian results in the main text instead of, or as well as, the frequentist, we prefer not to because some models that were non-significant in both the Bayesian and frequentist analyses had convergence issues in the blme package and analysis. As these models are non-significant, this does not affect the results or conclusions, but it does make reporting more cumbersome. We therefore feel it is more straightforward to present results generated based on typical GLMMs with which readers will be familiar with, while noting that the results of a complementary Bayesian analysis were essentially identical, increasing confidence in their reliability and trust that the warning message generated by the glmer function is not of concern.

Reviewer 3 also expressed concern regarding whether our random effects structure is too complex for the data. We note that incorporating ‘unnecessary’ random effects (i.e. those that are estimated near zero) is not statistically problematic. According to statistician Ben Bolker (https://bbolker.github.io/mixedmodels-misc/glmmFAQ.html#singular-models-random-effect-variances-estimated-as-zero-or-correlations-estimated-as---1): “if a variance component is zero, dropping it from the model will have no effect on any of the estimated quantities…conversely, if one chooses for philosophical grounds to retain these parameters, it won’t change any of the answers.”

3) Clarify the biological significance of the looking-time experimental results. The relationship of these results with species recognition processes was considered indirect at best and inferential. Specify what can be concluded once these results are clarified; is this still an impactful result? If so, emphasize that aspect, while also still discussing potential implications for species recognition. As the reviewing editor, I noticed that when you had more space to describe your results you were appropriately careful (yet this was not the case in several summary sentences). However, even then I think it would be better to be more direct about this limitation, while simultaneously celebrating the explicit knowledge that is obtained if warranted.

The issues raised by the editor and reviewers here are insightful and point to an interesting area of theory and terminological discussion within animal communication. Reviewer 2 states “it is unclear to me how the behavioral experiment can support any conclusion about species recognition”. We would first like to note that we did not use the term ‘species recognition’ in our manuscript, nor do we think that guenons, or any other primates ‘recognize species’ in the sense of categorizing animals based on species concepts as imposed by human researchers. In this research, we also do not attempt to infer the underlying neurological mechanisms involved in ‘recognizing’ objects or categories in a psychophysical sense, but rather are trying to identify signal components that lead to increased visual attention. We have opted for the term species ‘discrimination’, and use discrimination to refer simply to an ability to differentiate between two classes of stimuli. Our experiments represent a behavioral discrimination task and therefore do test this ability directly; we have updated the manuscript to clarify this (126-130).

Our experiments are designed to assess the types of facial traits that draw the attention of guenon subjects in a behavioral discrimination task. This is the first study to assess guenon responses to con- and heterospecific faces in this group, and this experimental component of our research is critical both to documenting guenon responses to these purported signals and to contextualizing computationally-derived results. Eye gaze is a commonly used metric for documenting visual attention and thereby inferring how subjects perceive the world (see review: Winters et al., 2015 Ethology “Perspectives: the looking time experimental paradigm in studies of animal visual perception and cognition”). When individual animals engage with their physical and social environment, they typically preferentially attend to things that are salient and of interest to them, such as predators, offspring, or mates. Our experiments are designed to use this tendency to understand what makes particular face patterns more interesting to guenons. We have clarified these research goals in our manuscript. In our experiments, the first condition implemented for both of our target species is one in which we pair a conspecific with a heterospecific. The results validate our experimental paradigm as they demonstrate that, consistent with evidence from other species, both species have a bias towards visual attention to conspecifics vs heterospecifics. The question in the subsequent experiments is then: how is this bias affected by manipulations of which facial pattern components are available? Our results are consistent with an interpretation that these face patterns are part of the perceptual experience and underlying sensory processing that is associated with the increased attentiveness to conspecifics over heterospecifics. We have edited the end of the Introduction and Materials and methods sections to be clear about the purpose of our experiments, and how we interpret our results.

We appreciate that our experimental results do not generalize to behaviors beyond eye gaze. Yet previous work has linked increased eye gaze to behavioral measures of mating interest in other primates (Shimojo et al., 2003; Dubuc et al., 2014, 2016), and we feel it is reasonable to suggest that conspecific-biased mate choice and resulting reproductive isolation may be associated with our observation of increased interest in conspecific faces in guenons. We have edited text in the discussion to present this conjecture with appropriate caution and to more explicitly present our logic in drawing parallels between eye gaze and potential mating behavior.

Finally, we have also made a number of additional changes to our manuscript based on other reviewer comments that were not included as “essential revisions” by the editor. These include: Moving descriptions of and details on a number of experimental methods related to the experimental population, stimuli, apparatus, and procedures from supplementary files to the core manuscript (Reviewer 1). We have also set in motion mechanisms to provide our experimental videos online and open access (Reviewer 1). Regarding being wild vs captive born, and the potential for exposure to heterospecifics, we assume that all animals have potentially been exposed to all potential heterospecifics, since some animals may have encountered heterospecifics either in captivity or in the wild (Reviewer 1). This does not affect any of our conditions, as aside from conspecifics all other conditions are generated from images of allopatric species. We have now added this information to the manuscript. We have clarified our language regarding eigenface scores and removed reference to it as a “classification scheme”. We have now referenced Santana et al.’s work, but note that this work is on facial complexity rather than facial distinctiveness (Reviewer 4). We have added a sentence about the face patterns of other primate radiations (Reviewer 4). We have also made a number of minor edits suggested by reviewer 4. In response to requests from the editorial support team, we have also updated the formatting of the figures and supplementary materials. All supplementary figures are now linked with a manuscript figure, and the text from the sections previously called Supplementary Methods and Supplementary Results has been moved to an Appendix.